# Interprofessional Collaboration in Fall Prevention: Insights from a Qualitative Study

**DOI:** 10.3390/ijerph191710477

**Published:** 2022-08-23

**Authors:** Isabel Baumann, Frank Wieber, Thomas Volken, Peter Rüesch, Andrea Glässel

**Affiliations:** 1Institute of Public Health, Zurich University of Applied Sciences (ZHAW), 8400 Winterthur, Switzerland; 2Center for the Interdisciplinary Study of Gerontology and Vulnerability, University of Geneva, 1205 Geneva, Switzerland; 3Department of Psychology, University of Konstanz, 78464 Konstanz, Germany; 4Institute of Biomedical Ethics and History of Medicine, University of Zurich, 8006 Zurich, Switzerland

**Keywords:** fall prevention, interprofessional collaboration, qualitative research, focus groups, community health services, older adults, evaluation, physical therapy, occupational therapy, general practitioners

## Abstract

(1) Background and objective: to explore the experiences of Swiss health care providers involved in a community fall prevention pilot project on barriers and facilitations in interprofessional cooperation between 2016 and 2017 in three regions of Switzerland. (2) Methods: semi-structured interviews with health care providers assessed their perspective on the evaluation of jointly developed tools for reporting fall risk, continuous training of the health care providers, sensitizing media campaigns, and others. (3) Results: One of the project’s strengths is the interprofessional continuous trainings. These trainings allowed the health care providers to extend their network of health care providers, which contributed to an improvement of fall prevention. Challenges of the project were that the standardization of the interprofessional collaboration required additional efforts. These efforts are time consuming and, for some categories of health care providers, not remunerated by the Swiss health care system. (4) Conclusions: On a micro and meso level, the results of the present study indicate that the involved health care providers strongly support interprofessional collaboration in fall prevention. However, time and financial constraints challenge the implementation. On a macro level, potential ways to strengthen interprofessional collaboration are a core element in fall prevention.

## 1. Introduction

### 1.1. The Risk of Falls among Older Adults

Falls are a major risk to the older adult’s health. A Cochrane review indicates that among people over 65 years, the fall incidence is about 33% per year and among people over the age of 85, it is about 50% per year [1]. Data from Switzerland for 2013 indicates about 90,000 falls among people over 65 years registered at the Swiss Council for Accident Prevention (BFU) [2]. Falls often lead to fractures, in particular proximal femoral factures. Besides the harm to the older adults who experience the falls, the subsequent treatment creates huge costs [3].

The risk of falls among older adults is multifactorial. Risk factors associated with falls include health-related factors such as poor nutrition [4], cognitive impairment [5,6,7,8], functional disabilities [6,9] and environmental factors such as home safety [10]. Due to this multifactorial nature of falls, different health care providers are involved in the prevention and the management of the fall risks [11,12]. Physical or occupational therapists carry out physical exercise programs or adjust home environments in a way to increase the older adults’ safety [12,13,14]. Nurses provide regular health care for the older adults in hospitals and homes and are thus particularly aware of their needs in terms of reducing fall risks [5]. If medication or similar treatments are applied, general practitioners are usually involved [15].

Therefore, preventing and managing falls calls for an interprofessional approach not only in inpatient but also in outpatient or community settings [16]. Interprofessional collaboration, defined as an active relationship between health care providers to work together to solve problems or provide services and to focus on a common goal, has been found to have mixed but predominantly positive effects on health care provided to community-dwelling older adults [17,18].

### 1.2. Interprofessional Fall Prevention

In conventional community-based fall prevention, health care providers often work independently from each other. For instance, a review study that focused on general fall prevention found that non-medical health care providers are at times not equally included in decision-making processes [19]. Interprofessional fall prevention, in contrast, allows health care providers to not only exchange experiences but also patient-related information to develop a tailored health care program and shared decision-making processes. A small qualitative study from Canada found that health care providers feel more “on top of things” and can learn from each other if they work together with providers from other disciplines [20].

This exchange of experiences and information is smoother if health care providers know each other personally [21]. Personal contacts allow reducing misunderstandings and ineffective care. In addition, the literature suggests that interprofessional collaboration requires specific communication skills, for instance, a common language [22,23]. Technical terminology including abbreviations or jargon of some professionals may not be understood by others [24]. Introducing a common language is thus important and needs mutual communication and exchange. However, this may be perceived as a loss of professional identity [24].

Another important factor for a smooth functioning of the interprofessional collaboration is role clarity, i.e., the definition of the function of each team member and her/his contribution to the collaboration. Due to potential role conflicts between the members of an interprofessional team, a structured protocol with predefined division of roles and responsibilities is helpful [25,26]. Such protocols have to be negotiated among the team members, a participatory process which is usually carried out in interprofessional meetings that allow to plan and coordinate health care [27]. If roles and responsibilities are not explicitly defined, experience shows that the team members feel overlooked or overburdened [21]. In contrast, if team members have an equal status or a perception of balance within the team, they trust each other more, which leads to a better functioning of the interprofessional team [28].

As a challenge, a review study that focused on older people living in the community found that interprofessional collaboration is more time-consuming than multi- or non-interprofessional collaboration, which is often not adequately reimbursed [26]. In particular, the coordination between the team members consumes large amounts of time [28]. Consequently, time pressure renders interprofessional collaboration challenging [22].

In support of the interprofessional collaboration, many of the studied interprofessional teams describe that they use specific tools for exchange of information about patients [25,29]. Some use electronic health records, which facilitate coordination in times of the digitalization of health care [26]. These tools allow team members to know about the patient’s health status and the other team members’ care activities [27].

### 1.3. Primary Prevention Program in Switzerland

In Switzerland, the foundation of health promotion Switzerland (“Gesundheitsförderung Schweiz”) launched the pilot project “Via Pilotprogramm Sturzprävention” between 2014 and 2016. The project aimed at improving fall prevention among community-dwelling older adults and particularly addressed older adults with a risk of falling or who have already fallen; it thus constitutes specific rather than general fall prevention. As part of this aim, it focused on improving the health care provision on fall prevention of older adults by building and strengthening the network between community health service providers based on standardizing interprofessional collaboration. The fall prevention pilot project was implemented in three regions in the German speaking part of Switzerland: two rural (regions 1 and 2) and one urban (region 3). The present study is the result of the qualitative research conducted in the context of a project evaluation.

The fall prevention pilot project followed a bottom-up approach. In each of the three different regions of Switzerland, it provided a tailored organizational structure to bring health care providers from different professions together to discuss and implement potential measures. Health care providers involved in this fall prevention pilot project were physicians (general practitioners (GP), geriatric specialists, and other specialists), physical therapists, occupational therapists, hospitals, home care providers, organizations of the civil society, and a senior citizens’ organization. An overview over the health professions involved is provided in Figure 1. In each region, health care providers and institutions developed their own measures, although partly shared and adopted each other’s measures. One measure consisted in organizing a continuous interprofessional training for health care providers in fall prevention. Another measure consisted in creating evidence-based tools (e.g., a registration form) to facilitate communication [30]. The interprofessional training included contents such as the recognition of stumbling blocks and the self-awareness of impaired vision in everyday life or exercises to train balance.

## 2. Materials and Methods

### 2.1. Study Design

In 2016, our research group was mandated by “Gesundheitsförderung Schweiz” to conduct a summative evaluation of the fall prevention pilot project. The design of our evaluation involved qualitative interviews and a focus group with several representatives of the five largest categories of health care providers, as well as a quantitative survey among all health care providers and representatives of the institutions involved in the fall prevention pilot project. The evaluation of the pilot project comprised six case studies: three in the pilot regions (regions 1 to 3) and three in comparison regions (regions 4 to 6). For each of the project regions, we included one comparison region with a similar socio-demographic structure and in a similar geographic setting (i.e., regions 4 and 5 are rural, region 6 is urban).

### 2.2. Methodology

In this paper, we focus on the results of the qualitative study. Therefore, the aim of our article is to explore the experiences of the health care providers involved in the fall prevention pilot project on barriers and facilitations in interprofessional cooperation in fall prevention in Switzerland. The results of the quantitative part of the evaluation have been presented in a report [31]. Qualitative methods offer the possibility to explore the research object from the perspective of those who experience an everyday relation to it, in the sense of the so-called user, consumer, or client perspective [32]. In contrast to the quantitative methodology, the qualitative approach promises greater openness to unexplored concepts or phenomena and focuses [33] on how people understand and interpret their social world [34].

### 2.3. Data Collection and Participants

Qualitative data were collected between September 2016 and March 2017 in individual telephone interviews and one focus group. In the pilot regions, participants were the members of the coordination group which was responsible for the implementation of the pilot project. The research team received a list of representatives of all occupations from the main representative responsible for the project. The researchers then contacted the representatives by email and telephone. In the control regions, the researchers browsed the internet to identify one representative of each occupation.

In each region, we aimed at interviewing one representative of the five largest categories of health care providers (physicians, physical and occupational therapists, home care nurses, and informal carers). In addition, a representative of a seniors’ organization participated in the focus group in region 1. While in the pilot regions (regions 1–3) all of the contacted representatives of the health care providers participated in the focus group or the individual interview, in the control regions (4–6) it was more difficult to recruit participants. For this reason, we interviewed four rather than five representatives in the control region, because some of the contacted health care providers were not available for an interview (see Table 1). This difference may bias the results in the direction that more support for interprofessional fall prevention may be observed in the pilot region.

Two trained researchers conducted the focus group. The two researchers who conducted the individual interviews and focus groups had no prior personal relationships with the interviewees. Researcher 1 moderated the focus group discussion, which was based on a semi-structured guide of questions focusing on the following six main topics: (1) collaboration between health care providers in fall prevention, (2) use and evaluation of tools and processes, (3) communication with the public, (4) use and evaluation of further training, (5) cooperation in the coordination group, (6) the perceived direction and strength of changes through the fall prevention pilot project, as well as (7) the possibility to comment on the topic. Researcher 2 composed the written protocol of the focus group. For reasons of quality assurance, the focus group was completed with a joint debriefing of the two researchers, in which the main results were consolidated. The qualitative interviews followed the same interview guide. The focus group and the qualitative interviews were audio recorded and transcribed verbatim based on the defined rules [35].

### 2.4. Data Analysis

The data were analyzed using a qualitative content analysis based on Mayring [36]. The aim of this analysis is to reduce information from the participants’ statements in such a way that the essential content remains and the structures and core characteristics become clearer. For this purpose, to develop a category system, an inductive procedure and a deductive procedure based on the six main topics of the semi-structured guide were used. The coding guide to create categories and the coding rules contained therein served as a basis for ensuring traceability and replicability of the procedure. The data analysis was managed by using the software MAXQDA version 12 (VERBI: Berlin, Germany). 

### 2.5. Quality Assurance of the Data Analysis

For reasons of transparency and comprehensibility, the participants received the scripted interviews for reading aloud and for making comments called “member checking” of the transcripts. Data were analyzed by researcher 2 using a code book and building up a system of written memos. Data were grounded in the text by using original quotations of the anonymized participants of the interviews. A researcher handbook was used for notes during the process. 

## 3. Results

Five main categories regarding interprofessional collaboration in fall prevention in Switzerland were identified in the study: Information and communication;Stakeholder and collaboration;Cooperation between the health care providers;Case-related cooperation;Continuous training measures.

In the following, we report the results for each category. We first focus on hindering aspects that were perceived by the health professionals before we present the facilitating aspects that were mentioned. 

### 3.1. Information and Communication

#### 3.1.1. Barriers

The registration forms that were developed to facilitate the communication between the health care providers are not equally used in all regions and by all categories of health care providers. In fact, the registration forms are particularly often used by home care providers. The following statements about the registration forms show excerpts of the status of use in the pilot regions. 

“I mean the registration sheets. So, they still have to prove themselves. It’s not quite like that yet.”(Region2_P1)

“From the registration forms I have received only two so far.”(Region2_P5)

“At the moment I’ve actually had only a few of these messages. And the messages we sent to the doctors didn’t come back. We didn’t hear anything more.”(Region2_P1)

#### 3.1.2. Facilitators

Nevertheless, in general, the health care providers in the pilot regions rated the familiarity of the registration sheets for assessing the risk of falls as equally high and rated the preparation of the registration sheets as good.

“The registration sheet has proved its worth to the extent that it is now known to all participants, and it is clear what happens with it. There is still development potential in the fact that general practitioners distribute concrete tasks to [home care]. From October to December, we sent 36 registration forms, eight of them were provided with the concrete feedback by the family doctor and 21 came back without anything being written, […] and for seven they did not receive patient’s consent to be sent in.”(Region3_P4)

In the three comparison regions, the health care providers were interviewed in the context of the qualitative interviews to assess their perceptions on the usefulness of the so-called registration form for their work. The structure and content of the registration form were briefly explained. In principle, the players in the comparison regions were open minded towards such a registration sheet and considered it to be useful.

### 3.2. Stakeholder and Collaboration

#### 3.2.1. Barriers

In the interviews, it became clear that fall prevention as an isolated topic of cooperation is not the starting point, but it is considered in the context of more complex case-related questions of cooperation in care.

“One difficulty of the whole interview is that it focuses on the fall. We as family doctors and home care practices a comprehensive care and often fall is a topic among many. And this makes it a bit difficult to simply ask about falls. That is of course a topic. When I then talk to home care, it’s a topic, how can you support. And in the palliative situation, of course, the patient is even worse ‘weighed up’ […], do you need a round table with the relatives, for example? This is about bringing all professional groups together at one table. Why we don’t do that is also the time. A round table is simply very time-consuming. It needs all the players, and you can only do that in individual cases.”(Region6_P2)

The health care providers in the comparison regions also report existing networking and cooperation structures. They originate in other areas of health care, such as neurology or palliative care. These structures benefit the purpose of fall prevention. Cooperation usually refers to an exchange of information, which can take place orally at a round table, by telephone, in writing by e-mail, or in the form of reports. However, it is reported to not take place regularly or systematically and is still not paid or reimbursed by the Swiss health care system.

“There is little cooperation, and in most cases, this is an exchange of information but not effective cooperation. So, the information about medication, about diagnosis list, like for example with Parkinson’s [disease], which also per se in the diagnosis contains a fall potential, or that one has prescribed physical therapy for this or that.”(Region6_P2)

“No, and it’s more for patients, too, in the palliative situation. So, it is not primarily about falls. But the round tables are rather for the overall situation: Is it still possible at home, what can you do with home care? And then the risk of falling is perhaps a sub-theme. […]. But there is practically no effective, coordinated cooperation only about falls.”(Region6_P2)

#### 3.2.2. Facilitators

In contrast, continuous training measures had a positive effect on networking and thus on cooperation between the actors in the pilot regions.

“You then get to know the people who work in the same city, in the same place. So, I felt that this was very positive […] and the exchange can take place.”(Region3_P2)

“I think it’s very important to bring everyone to one table so that the network can develop. And this is only possible through personal contacts, […] then it is easier to make contact. It is always positive to meet people from other professions not only for fall prevention. This also has positive effects on other things. If you know the people personally, then the threshold is lower to call or ask.”(Region2_P1)

“It’s always good to know the people [health care professionals] who are in contact with the general practitioner.”(Region2_P4)

Personal contacts allowed removing interpersonal barriers, which facilitated queries and had a positive effect on the regional inter-professional cooperation. Specific details relevant to the health care provisioning can be exchanged more easily.

### 3.3. Cooperation between the Health Care Providers

#### 3.3.1. Barriers

In one of the pilot regions, the project has led to little noticeable change from the point of view of doctors and occupational therapists. Nevertheless, contact was facilitated:

“We now know each other’s face and thus the threshold for a telephone call is lowered.”(Region3_P3)

In another pilot region, the project has hardly brought about any changes in the day-to-day practice of doctors, albeit in feedback. The cooperation with the core actors (physiotherapy, occupational therapy, hospital, and home care) has remained the same. However, the involvement of the optician has been positive and thus an increase in interprofessional collaboration is discernible.

“I have made positive experiences concerning the cooperation with the general practitioners, just by sending the registration form and the prescription actually came back, otherwise with the other instances [health care providers] the networking, at least in my environment, has not taken place.”(Region3_P3)

In the comparison regions, the cooperation between the health care providers revealed a similar pattern as in the pilot regions. Certain professions are in closer contact with each other than others, such as physicians and physical therapists, or physicians and home care providers.

“Physical therapists who go home. We now also have occupational therapy in the region, which I have also referred patients to in rare cases. First and foremost, physical therapy and home care.”(Region5_P5)

“We have very little contact with occupational therapy etc. We have now for the first time, the [nonprofit organization in health care] has now made such a campaign, where they offer inexpensive fall prevention measures. And there we came into contact for the first time.”(Region6_P3)

#### 3.3.2. Facilitators

In some of the pilot regions, the interprofessional fall prevention was assessed more positively.

“It has been interesting, watching different foci. And it certainly contributed positively to the cooperation.”(Region2_P1)

Certain health care providers work more closely together, such as physicians and physical therapy, or physicians and home care. First networks are growing.

“That’s why only [home care] and general practitioners work interprofessional now. So, we know from each other, and everyone has his share of what he perceives, but overall, the topic of fall prevention is still less important that other topics.”(Region3_P4)

Case-related exchange and interprofessional cooperation are fostered by spatial proximity, for instance by being in the same village or in the same building. Here, group medical practices have an advantage over individual medical practices in using this potential for cooperation. An occupational therapist from a comparison region states:

“When patients come to us and they need physical and occupational therapists, they usually have both. This means that I am not at all in need of finding the telephone number of another physical therapist.”(Occupational therapy_P4)

Coordinated communication between the health care providers is highly important. Tools such as the registration form, for instance, had a positive effect on collaboration between the general practitioners and the physical therapists, leading to prescriptions and follow-up prescriptions.

### 3.4. Case-Related Cooperation

#### 3.4.1. Barriers

Skeptical assessments of case-related cooperation and networking have become apparent, which relate to existing framework conditions and the available time resources. The actors expressed the need for exchange:

“The physical therapist or even the general practitioner simply have no time, or rarely, to seek exchange. In their professional environment, where in both professions almost every tenth min has to be accounted for, I don’t think that [cooperation] has much room.”(Region3_P2)

“Not much has changed, we have always had a good cooperation between physical therapy, occupational therapy, hospital, home care and ourselves […] If possible, we should prefer to do something electronically. The time resources of all participants are limited.”(Region2_P5)

“Home care, we are dependent on working together with others. General practitioners prescribe home care, but general practitioners do not seek this form of interdisciplinary cooperation. It is a historical structure. This hierarchical thinking is certainly a hindrance. The only thing where this begins to dissolve is in palliative care, where general practitioners or doctors and nurses in psychiatry are dependent on each other. Is there an exchange […], which significantly increases patient satisfaction and improves care when working together on an interdisciplinary basis? But there is little time for this.”(Region3_P6)

#### 3.4.2. Facilitators

However, there are also positive experiences with case-related cooperation. Case-related exchange between physical and occupational therapy is sometimes created without being explicitly triggered by fall prevention.

“The cooperation with physical therapists and occupational therapists is not structured at all. But we know that there are some [health care providers] who make similar efforts.”(Region3_P6)

### 3.5. Continuous Training Measures

Most of the participants in the pilot regions responded positively to the continuous education training courses on fall prevention. The main reasons were that they enabled health care professionals to get to know each other and to establish networks. In addition, the courses employed an interdisciplinary approach to fall prevention.

#### 3.5.1. Barriers

However, the continuous trainings offered only limited space to break down barriers for future collaboration or interprofessional exchange.

“I missed this opportunity to use it for an interdisciplinary meeting. They were physiotherapists and occupational therapists. But there was no moment when one could have exchanged ideas. I have already reported this.”(Region3_P6)

In the comparison regions, continuing training about fall prevention was offered to individual professional groups in the past, for example in physiotherapy. Interprofessional continuing trainings were rarely attended.

“One could certainly do [interprofessional continuous training] on the field of fall prevention. But sometimes interprofessional continuous training is really difficult, because health care professionals bring along different conditions and have different needs.”(Region6_P2)

#### 3.5.2. Facilitators

Continuous training measures in the pilot regions had a positive effect on the cooperation between health care providers in these regions. Personal contact, getting to know each other, and the removal of barriers to making contact for queries facilitated interprofessional collaboration.

“You then get to know the people who work in the same city, in the same place. Well, I found that very positive […] and the exchange can take place.”(Region3_P2)

“I think it’s very important to bring everyone together at one table so that the network can develop. And this is only possible through personal contacts, if you know who is behind it, it is easier to make contact. It is always positive to meet people from other professions not only for fall prevention. This also has positive effects on other things. If you know the people personally, then the threshold to call or ask is lower.”(Region2_P1)

“It’s always good to know the people who have to do with the general practitioner and physical therapy is also a topic that runs parallel where [the home care organization] doesn’t notice anything.”(Region2_P4)

Content-related aspects that were perceived positively by the participants were:

“Positive impressions of the continuous training, which was simply clearly practice-oriented and caused aha-effects with the employees.”(Region3_P6)

“Also, different workshops. One was with the optician, so this self-awareness, what it can mean with bad visibility, what effects it can have. That has already triggered something.”(Region2_P4)

In conclusion, we can maintain that the qualitative interviews provide in-depth insights about the desirability of cooperation and obstacles in the feasibility of interprofessional collaboration in fall prevention. Our analysis highlights that cooperation between certain occupational groups—e.g., for all professions with home care has improved, whereas for others not much has changed through the project—e.g., between general practitioners and physical therapists.

## 4. Discussion

The present article aimed to explore how health care providers involved in a pilot project evaluate the impact of the project on the effectiveness of fall prevention in older adults in Switzerland regarding facilitators and barriers of interprofessional collaboration for building a professional network. Our study contributes to the literature by providing evidence on interprofessional collaboration in fall prevention, a field of study that has received little attention in Switzerland.

Three statements were evaluated as particularly relevant by most health professionals: First, interprofessional continuous trainings were highly appreciated. On the one hand, the trainings allowed health professionals to establish relevant networks. On the other hand, health professionals perceived improved quality of care due to an interdisciplinary approach to fall prevention. These findings are in line with previous research from the United States that emphasized how interprofessional education improved fall prevention by expanding health professional’s disciplinary view and by improving communication between the disciplines [37,38].

Second, interprofessional collaboration was rated as resource intensive. To foster interprofessional collaboration in fall prevention, more time, financial resources, and/or more health care providers are needed. Previous research shows that a lack of human resources and financial compensation constitute serious barriers to interprofessional fall prevention programs [39,40,41]. A study from Switzerland highlighted that in the outpatient setting, remuneration for some tasks in interprofessional collaboration is provided only for some but not all health care professionals [39].

Third, defining the roles of health care professionals was evaluated as a major challenge for successful interprofessional fall prevention. These findings complement earlier research [38,39,42]. For instance, a study from Australia that was carried out in a similar setting as our study showed that overlapping roles led to a sense of competition and even rivalry between the professionals rather than to improved interprofessional collaboration, particularly if they competed for business [42].

### Limitations

First, the present study employed a post hoc design, in which interviews in comparison regions served to provide insights into how the actual fall prevention practice in pilot regions might differ. However, this design does not allow drawing strong conclusions about the size and causality of the observed differences and similarities. Second, the small number of interviews and focus groups prevented the qualitative analyses form attaining a high level of saturation, in which the content and structure derived from earlier interviews and focus groups would have been confirmed and no new arguments were added by the later interviews and focus groups. The first and second limitations both may lead to results affected by information bias. Information bias arises if the data collected systematically deviate from the truth [43,44]. Such bias may be present in our data in particular for two reasons. First, respondents of the pilot study may have had an interest to report their work in a more positive light than they experienced because they may feel responsible for the implementation of the pilot project and interested in the continuation of the project. Second, participants in the focus group may have mutually influenced each other. The first source of bias was addressed in our study by using a control group. The second source of bias was partially addressed by using individual interviews.

A third limitation of the project was that the fall prevention pilot project was implemented in regions that had already pre-existing interprofessional networks. Thus, the transferability to other regions might be limited such that the implementation of the pilot project in regions without such interprofessional networks might turn out to be more difficult. In general, it should be noted that all interviews were conducted in the German-speaking regions of Switzerland and the results are not easily transferable to the French and Italian-speaking regions of Switzerland. A final limitation of the project is that the data stems from 2017 and is thus not very recent. Nevertheless, we consider our study as an important contribution to the field of fall prevention in Switzerland for two reasons. First, falls constitute the deadliest cause of all non-occupational injuries in Switzerland [45]. There is thus a need to better understand how fall prevention may be best addressed. Second, to our knowledge, besides one other project [46], the pilot project studied here is one of the only fall prevention programs that has been evaluated recently. In contrast to the other project mentioned, our study includes a control group (i.e., health professionals in the control regions) which allows to better capture a potential causal effect of the fall prevention program. Our study therefore fills a gap in the knowledge on fall prevention in the Swiss context.

## 5. Conclusions

With the increasing number of older adults in Switzerland as well as the continuously increasing life expectancy, the incident rates of falls are likely to increase. Given the often-severe consequences of falls in terms of individuals’ health, fall prevention is a central topic from a health care as well as a health economic perspective. Although there is evidence on a micro and meso level demonstrating that fall prevention measures can improve older adults’ health, quality of life, and independence, there is the need to complement these advances by improving the interprofessional collaboration in fall prevention on the macro level in the sense of public health policy.

## Figures and Tables

**Figure 1 ijerph-19-10477-f001:**
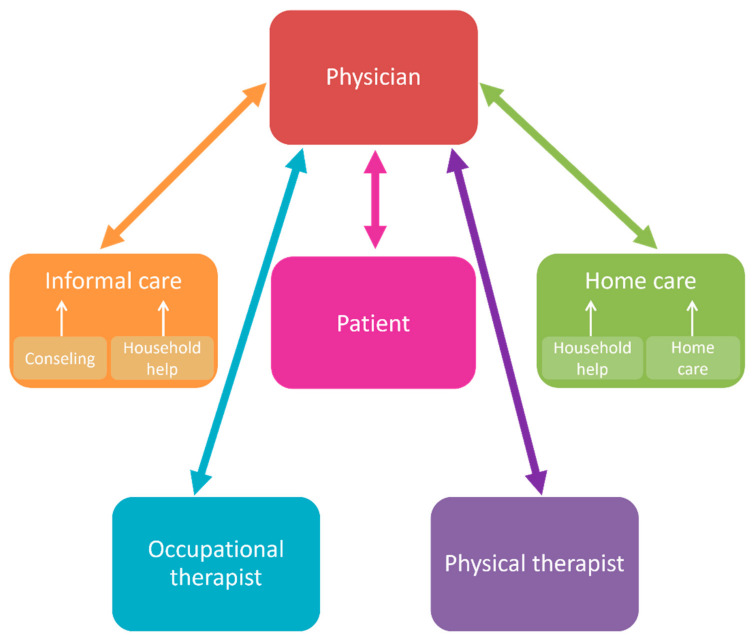
Overview of the health professions involved in the Swiss primary prevention program.

**Table 1 ijerph-19-10477-t001:** Overview of the participants in the focus group and interviews (note: NA—no answer).

Region	Organization/Occupation	Education	Year of Birth	Overall Occupational Experience (in Years)	Occupational Experience in Fall Prevention (in Years)
1 (pilot, rural)	Physician	GP	1970	-	6
1 (pilot, rural)	Occupational therapist	Occupational therapy HF	1978	-	10
1 (pilot, rural)	Occupational therapist	Occupational therapy HF	1967	-	15
1 (pilot, rural)	Physical therapist	Physical therapy	1964	-	3
1 (pilot, rural)	Informal care	Nursing sciences HF	1965	-	3
1 (pilot, rural)	Senior citizen	Apprenticeship	1939	-	3
2 (pilot, rural)	Physician	GP	1967	17	1
2 (pilot, rural)	Occupational therapist	BSc in Occupational therapy	1974	18	18
2 (pilot, rural)	Physical therapist	Physical therapy	1975	26	26
2 (pilot, rural)	Home care nurses	Nursing expert	1967	25	8
2 (pilot, rural)	Informal care	Adult sports trainer	1963	14	5
3 (pilot, urban)	Physician	GP	1965	23	20
3 (pilot, urban)	Physical therapist	Physical therapy	1988	NA	NA
3 (pilot, urban)	Occupational therapist	Occupational therapy HF	1961	25	13
3 (pilot, urban)	Home care nurses	Nursing sciences	1973	20	13
3 (pilot, urban)	Informal care	Adult sports trainer	1963	13	5
4 (control, rural)	Occupational therapist	Occupational therapy	1972	15	5
4 (control, rural)	Home care nurses	Nursing sciences	1992	3	2
4 (control, rural)	Physical therapist	Physical therapy	1958	36	27
4 (control, rural)	Physician	GP	1971	19	19
5 (control, rural)	Occupational therapist	Occupational therapy	1970	21	21
5 (control, rural)	Physician	GP	1954	38	38
5 (control, rural)	Informal care	NA	1962	32	11
5 (control, rural)	Physical therapist	Physical therapy	1961	30	30
6 (control, urban)	Physical therapist	Physical therapy	1953	39	2
6 (control, urban)	Physician	GP	1968	23	11
6 (control, urban)	Occupational therapist	Occupational therapy	1969	16	22
6 (control, urban)	Home care nurses	Nursing sciences	1984	8	3

## Data Availability

Requests to access the data can be addressed to ZHAW and Gesundheitsförderung Schweiz.

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
