# Peer review of "Interprofessional Collaboration in Fall Prevention: Insights from a Qualitative Study"

_ijerph, 2022, doi:10.3390/ijerph191710477_

Round 1

Reviewer 1 Report

Despite the fact that "its design does not allow firm conclusions to be drawn about the size and causality of the differences and similarities observed", the study concludes (in line with some quotes exposed in the introduction) "that the involved health care providers strongly support interprofessional collaboration in fall prevention". Given that it is a qualitative study with a low number of interviews, which "prevented the qualitative analyzes from reaching a high level of saturation" I suggest a more detailed discussion of potential information biases.

The authors declare that they have no conflicts of interest and express their gratitude to all the health care providers who participated in our interviews. In the previous line of discussion, I suggest some comment on the possible relationship, beyond the personal, between the components of the research group and the service providers in the pilot study.

Finally, I would like to know if the time elapsed between the collection of the data/interviews and the presentation of the article could influence the results of the study in some way.

Reviewer 2 Report

Thank you for the opportunity to review this paper.  It is interesting and has merit.  I would like to see a section regarding rigour of the study, although the discussion does comment on issues such as data saturation etc.  Recruitment was not clear, nor was how some potential participants refusing to be interviewed commented on as a limitation/source of bias.

Round 2

Reviewer 1 Report

N

Reviewer 2 Report

Feedback addressed satisfactorily